# Molecular Landscape of the Coagulome of Oral Squamous Cell Carcinoma

**DOI:** 10.3390/cancers14020460

**Published:** 2022-01-17

**Authors:** Marine Lottin, Simon Soudet, Julie Fercot, Floriane Racine, Julien Demagny, Jérémie Bettoni, Denis Chatelain, Marie-Antoinette Sevestre, Youcef Mammeri, Michele Lamuraglia, Antoine Galmiche, Zuzana Saidak

**Affiliations:** 1EA7516 CHIMERE, Université de Picardie Jules Verne, 80054 Amiens, France; Lottin.Marine@chu-amiens.fr (M.L.); Soudet.Simon@chu-amiens.fr (S.S.); julie.fercot@etud.u-picardie.fr (J.F.); floriane.racine@etud.u-picardie.fr (F.R.); Bettoni.Jeremie@chu-amiens.fr (J.B.); Chatelain.Denis@chu-amiens.fr (D.C.); sevestre.marie-antoinette@chu-amiens.fr (M.-A.S.); 2Department of Biochemistry, Center for Human Biology, Amiens University Hospital, 80054 Amiens, France; 3Department of Vascular Medecine, Amiens University Hospital, 80054 Amiens, France; 4Department of Hematology, Center for Human Biology, Amiens University Hospital, 80054 Amiens, France; Demagny.Julien@chu-amiens.fr; 5Department of Maxillofacial Surgery, Amiens University Hospital, 80054 Amiens, France; 6Department of Pathology, Amiens University Hospital, 80054 Amiens, France; 7Laboratoire Amiénois de Mathématique Fondamentale et Appliquée (LAMFA), CNRS UMR7352, Université de Picardie Jules Verne, 80069 Amiens, France; youcef.mammeri@u-picardie.fr; 8Department of Oncology, Amiens University Hospital, 80054 Amiens, France; Lamuraglia.Michele@chu-amiens.fr

**Keywords:** tumor coagulome, OSCC, tumor microenvironment, immune checkpoints

## Abstract

**Simple Summary:**

Cancer is associated with a wide spectrum of hemostatic complications that range from thrombotic events to hemorrhage. The tumor coagulomes, i.e., the essential actors that locally regulate coagulation and fibrinolysis, play a key role in these complications. They might also play a regulatory role in various cell types of the tumor microenvironment. Here, we explored the coagulome of Oral Squamous Cell Carcinoma (OSCC) across tumor types, between OSCC tumors and within individual tumors. The coagulome of OSCC is characterized by a high expression of antipodal activators of coagulation and fibrinolysis, and subpopulations of pro-coagulant and pro-fibrinolytic cancer cells coexist within individual tumors. Importantly, we noted that dendritic cells within OSCC with a procoagulant profile express high levels of key immune checkpoint molecules. Further studies examining a possible negative modulation of the tumor’s adaptive immune response by the coagulation process are warranted.

**Abstract:**

Background: Hemostatic complications, ranging from thromboembolism to bleeding, are a significant source of morbidity and mortality in cancer patients. The tumor coagulome represents the multiple genes and proteins that locally contribute to the equilibrium between coagulation and fibrinolysis. We aimed to study the coagulome of Oral Squamous Cell Carcinoma (OSCC) and examine its link to the tumor microenvironment (TME). Methods: We used data from bulk tumor DNA/RNA-seq (The Cancer Genome Atlas), single-cell RNA-seq data and OSCC cells in culture. Results: Among all tumor types, OSCC was identified as the tumor with the highest mRNA expression levels of *F3* (Tissue Factor, TF) and *PLAU* (urokinase type-plasminogen activator, uPA). Great inter- and intra-tumor heterogeneity were observed. Single-cell analyses showed the coexistence of subpopulations of pro-coagulant and pro-fibrinolytic cancer cells within individual tumors. Interestingly, OSCC with high *F3* expressed higher levels of the key immune checkpoint molecules *CD274*/PD-L1, *PDCD1LG2*/PD-L2 and CD80, especially in tumor dendritic cells. In vitro studies confirmed the particularity of the OSCC coagulome and suggested that thrombin exerts indirect effects on OSCC cells. Conclusions: OSCC presents a specific coagulome. Further studies examining a possible negative modulation of the tumor’s adaptive immune response by the coagulation process are warranted.

## 1. Introduction

A broad spectrum of hemostatic complications, ranging from thromboembolism to hemorrhage, represent a significant source of morbidity and mortality in cancer patients [1,2]. Oral squamous cell carcinoma (OSCC) represents the most frequent histological form of tumor of the oral cavity and arises in the context of chronic exposure to tobacco and alcohol [3]. Patients with OSCC receive surgical procedures and medical treatments that put them at risk of thrombosis and hemorrhage [4,5,6], but they are usually considered to be at low risk of systemic thrombosis [7,8]. Understanding the biological basis for patient susceptibility to this large spectrum of complications requires fundamental studies on tumor physiology. 

The concept of the tumor coagulome refers to the equilibrium that exists between coagulation and fibrinolysis within tumors, and the contribution of multiple genes and proteins to this equilibrium [9]. Over the past decade, systems biology approaches, especially using genomics, have been applied to study the «core» components of the tumor coagulome, i.e., the local regulation of the main effectors of coagulation [10,11,12,13]. Clot formation is induced by the polymerization of fibrin resulting from thrombin activity, and it is initiated by the Tissue Factor (TF) encoded by the gene *F3* [14]. The overexpression of *F3* is a pivotal event in the establishment of the hypercoagulable state that characterizes most human tumors [14]. Conversely, fibrinolysis, the process that leads to clot destruction, depends on the activation of plasmin, involving the plasminogen activators uPA and tPA encoded by the genes *PLAU* and *PLAT*, respectively. In addition to the regulation of clot formation, the protease thrombin can also have a direct effect on cancer cells and the cells of the tumor microenvironment (TME) through its ability to interact with the protease-activated receptor-1 (PAR-1), a G-protein coupled receptor encoded by the gene *F2R*. The interaction of thrombin with PAR-1 modulates the growth of cancer cells and exerts complex effects on the malignant phenotype [15]. The serine protease uPA interacts with the receptor uPAR (urokinase-type plasminogen activator receptor, encoded by the gene *PLAUR*). Together with the serpin inhibitor plasminogen activator inhibitor-1 (PAI-1, encoded by the gene *SERPINE1*), uPA and uPAR are well-established markers of tumor prognosis [16]. 

Compared to other primary tumors, Head and Neck Squamous Cell Carcinoma (HNSCC), including OSCC, express high levels of pro-coagulant (*F3*) and pro-fibrinolytic (*PLAU*) genes [13]. The co-expression of high levels of pro-coagulant and pro-fibrinolytic genes establishes OSCC as an interesting model. Why OSCC may have such a specific coagulome is presently unclear as other sublocations of HNSCC, such as oropharyngeal tumors that arise in the context of chronic infection with oncogenic Human Papillomavirus (HPV), are characterized by low mRNA levels of *F3* and *PLAU* [17]. Interestingly, the oral cavity represents a specific biological environment with unique properties, such as a remarkable healing ability [18]. Coagulation contributes to wound healing and to the installation of the phenomenon of a «wound that does not heal» that characterizes most tumors [19]. With this in mind, a systems-level exploration of the coagulome of OSCC is interesting from both a medical and a fundamental perspective. 

The tumor microenvironment (TME) represents the non-malignant cells and structures (e.g., extracellular matrix) that are found within tumors [20,21]. The various cellular components of the TME, including cancer-associated fibroblasts, vascular cells and immune cells of myeloid or lymphoid lineages, can play a pro- or anti-tumor role depending on the context [20,21]. The TME is not only increasingly recognized as a critical determinant of tumor progression but is also emerging as a therapeutic target, as recently shown by the advent of immune checkpoint blockers [22]. A key checkpoint of the adaptive immune response against tumors is the interaction between the programmed cell death protein-1 (PD-1) molecule and its ligands PD-L1 and PD-L2 encoded by the *CD274* and *PDCD1LG2* genes, respectively. The expression of PD-L1 or PD-L2 on the surface of cancer cells and antigen-presenting cells induces a state of anergy in T cells. Targeting this interaction with clinically approved molecules was shown to be efficacious against recurrent/metastatic stages of HNSCC in phase 3 trials [22]. A second important immune checkpoint that can be targeted for therapeutic purposes depends on the interaction between Cytotoxic T-Lymphocyte Associated Protein 4 (CTLA4) and its ligand CD80. Understanding in which context these checkpoints are involved is an important research objective in order to optimize the efficacy of immune checkpoint inhibitors (ICI) and identify biomarkers for therapeutic stratification [23].

Importantly, a number of studies suggest the existence of an interplay between the coagulation system and the TME. The coagulation system has a potent effect on myeloid cells and initiates inflammation during the physiological healing process [24]. In the TME, coagulation might contribute to the recruitment of cells of the myeloid/lymphoid lineages and modulate their activation and function [25,26]. In the present study, we aimed to explore the regulation of the coagulome of OSCC and its link to the TME of these tumors. 

## 2. Materials and Methods

### 2.1. Bulk RNA-Seq and DNA Methylation Analysis from Human Tumors

RNA-seq is a powerful method for transcriptomic analyses. The results of bulk RNA sequencing of tumors can be obtained from a number of freely available resources, such as The Cancer Genome Atlas (TCGA). In this study, we retrieved gene expression and methylation data from TCGA [27] using cbioportal: https://cbioportal.org (accessed on 1 October 2021). Gene expression was given as RSEM (RNA-Seq by Expectation-Maximization), an algorithm used to estimate the gene expression based on the number of sequencing reads, after correcting for potential bias. HumanMethylation450 is an array-based method that permits the quantification of methylation of 450,000 CpG dinucleotide (96% of CpG islands and 92% of CpG shores are represented by at least one probe). HM450 beta values were retrieved from cBioPortal and used for the analysis (0 = unmethylated, 1 = full methylated). The data on gene mutations and copy-number alterations were retrieved from cBioPortal (TCGA cohort) for the genes *F3*, *PLAU*, *PLAUR*, *PLAT*, *SERPINE1* and *F2R*. The clinical characteristics of the 321 OSCC tumors available in TCGA are summarized in Appendix A.

### 2.2. Single-Cell Transcriptomics

Compared to tumor bulk RNA-sequencing, single-cell RNA-sequencing (scRNA-seq) permits cell type-specific analyses and provides access to intra-tumor heterogeneity. Single-cell RNA-seq data from 5902 cells were obtained from GSE103322 [28]. Pre-processing and quality control of the scRNA-seq data are described in detail in Puram et al. [28]. The data include 2215 malignant cells obtained from 18 HPV^-ve^ OSCC. The data on a sufficiently high number of cancer cells were available for further analysis for 10 of these tumors. 

### 2.3. Tumor Microenvironment Analysis

The ESTIMATE method uses gene expression signatures to infer the fraction of stromal (i.e., fibroblastic/mesenchymal cells) and immune cells in tumor samples and is described in [29]. The ESTIMATE score is based on two gene signatures (‘immune signature’ and ‘stromal signature’). The genes associated with the immune signature were identified using leukocyte methylation scores, shown to correlate with the presence of leukocytes. The genes present in the stromal signature were selected by comparing non-hematopoiesis genes in the tumor cell fraction and matched the stromal cell fraction in different cancer data sets. More detail about the construction of these gene scores can be found in Yoshihara et al. [29]. 

### 2.4. Gene Set Enrichment Analysis (GSEA)

GSEA, a computational method used to determine whether a set of genes shows statistically significant, concordant differences between two biological states, was performed using the Java GSEA desktop application (https://www.gsea-msigdb.org/gsea/index.jsp, accessed on 1 October 2021). We used curated hallmark gene sets, downloaded from the GSEA website, to compute their over-representation in RNA-seq samples expressing *F3* vs. *PLAU* in cancer cells from tumor #20 from GSE103322. The analyses were conducted using 1000 permutations [30].

### 2.5. Cell Culture and Reagents

The human OSCC cell lines PECA/PJ34, PECA/PJ41 and SCC9 were purchased from LGC Standards (Strasbourg, France) and compared to three human glioma cell lines (A172, U118MG and SW1088). Cell lines were authenticated using short tandem repeat profiling (LGC Standards, Strasbourg, France) and cell cultures were routinely checked for mycoplasma contamination. All cell lines were cultured in Dulbecco Modified Eagle’s Medium (DMEM, purchased from Sigma Aldrich (Saint-Quentin Falavier, France) supplemented with 5% fetal calf serum, 1% added glutamine and antibiotics (all from Sigma Aldrich). Thrombin and dabigatran were purchased from Sigma Aldrich. Human Interferon-γ was purchased from R&D Systems (Minneapolis, MI, USA). The growth of cancer cells was monitored after one rinse with PBS, a step of fixation with methanol and staining with crystal violet (Sigma Aldrich, Saint Quentin Falavier, France). After a wash step, the crystal violet dye was solubilized and measured by absorbance at 570 nm. The results were expressed as % of control.

### 2.6. Antibodies and Immunoblot Analysis

The procedures used for immunoblot analysis are reported in detail elsewhere [31]. The primary antibodies used in this study were: anti-TF (TF9-10H10, Sigma Aldrich), anti-uPA (Ab169754, Abcam, Paris, France), anti-tPA (Ab157469, Abcam), anti-PAI-1 (Ab66705, Abcam), anti-PAR-1 (E9J9L Rabbit mAb, #79109, Cell Signaling Technology, Danvers, MA, USA), anti-PD-L1 (Ab213524, Abcam) and anti-β-actin (A5441, Sigma-Aldrich). For the original Western blots, see Appendix A.

### 2.7. Statistical Analyses

Comparisons of two groups of numeric data were performed using the unpaired Wilcoxon–Mann–Whitney test. In all analyses, the Bonferroni correction was applied to control for multiple testing. Kaplan Meier analyses and the log-rank test were performed to compare the overall survival (OS) and disease-free survival (DFS) between OSCC tumors with high *F3* (>median) vs. low *F3* expression (<median), high *PLAU* (>median) vs. low *PLAU* expression (<median) and high *PLAT* (>median) vs. low *PLAT* expression (<median). *p* < 0.05 was set as the threshold for significance. All statistical analyses were performed with R version 4.1.0 (https://www.r-project.org, accessed on 1 November 2021). Correlation analyses were carried out using packages Hmisc and corrplot, calculating Spearman correlation coefficients (r).

## 3. Results

### 3.1. OSCC Express the Highest Levels of F3/PLAU across Human Tumors

We examined the mRNA levels of six genes that constitute the «core» tumor coagulome: *F3*, *PLAU*, *PLAT*, *PLAUR*, *F2R* and *SERPINE1* encoding TF, uPA, tPA, uPAR, PAR-1 and PAI-1, respectively, in tumors from TCGA (Figure 1). A pan-cancer comparison revealed great differences among the different types of primary tumors. Strikingly, OSCC was the human tumor type with the highest expression levels of *F3* and *PLAU* (Figure 1). However, great heterogeneity in *F3/PLAU* expression was apparent across individual OSCC tumors. The high expression of *F3* and *PLAU* and the tumor heterogeneity were evident from the direct comparison of OSCC and *Glioblastoma multiforme* (GBM), a tumor type associated with one of the highest risks of thromboembolic events (Figure 2).

We examined the extent to which the coagulome of OSCC is correlated with tumor staging (TNM status) and histological grade (Appendix A). We found no significant differences in *F3*, *PLAU*, *PLAT* mRNA expressions according to the T or N status. A small yet statistically significant difference was noted in *PLAU* mRNA expression depending on tumor grade, with higher *PLAU* levels in less differentiated tumors (G2/3) (Appendix A). Tobacco history and patient age were also considered as potentially interesting variables linked to the coagulome of OSCC, but no significant association was found between the expression of the coagulome genes and patient age or number of pack years (Appendix A). Further examination of *F3*/*PLAU* expression did not show any significant link to overall survival or disease-free survival of OSCC patients, suggesting that the coagulome is not strongly related to OSCC progression (Appendix A). 

### 3.2. The Coagulome of OSCC Is Genetically Stable and Is Correlated with Locus-Specific DNA Methylation

To address the mechanisms that regulate the coagulome, we searched for gene amplifications/deletions/mutations involving the genes of the coagulome in TCGA. The search identified the gene amplifications of *PLAT* and *SERPINE1* as the most frequent genomic events in OSCC, albeit at a low frequency of 2.2% (Appendix A). Somatic tumor mutations in *STK11*, *KRAS*, *CTNNB1*, *KEAP1*, *CDKN2B* and *MET*, which were reported to be associated with cancer-induced venous thromboembolism in an independent study [12], were infrequent in OSCC. We found no statistically significant association of any of the corresponding mutations with the expression of the coagulome in TCGA (data not shown). Meanwhile, there was great variation in locus-specific DNA methylation across tumors: a strong negative correlation (Spearman r = −0.65, *p* = 7.72 × 10^−40^) was found between *F3* mRNA expression and *F3* gene methylation. A slightly less strong yet clear negative correlation between mRNA expression and gene methylation was also observed for *PLAU* and *PLAT* (r = −0.44, *p* = 1.93 × 10^−16^ and r = −0.42, *p* = 7.85 × 10^−15^, respectively) (Figure 3). We concluded that the coagulome of OSCC is overall genetically stable and that its expression is correlated with locus-specific DNA methylation

### 3.3. The Coagulome of OSCC at the Single-Cell Resolution

Bulk RNA-seq data reflect gene expression in all cell types, including tumor cells and cells of the TME. We used scRNA-seq data from Puram et al. [28] to analyze the relative expression of each actor of the coagulome in different cell types, as shown in Figure 4 and Figure 5. In OSCC, on average, cancer cells expressed high mRNA levels of *F3*, *PLAU* and *PLAT*, but little if any of *PLAUR*, *F2R* or *SERPINE1*. Conversely, tumor-associated macrophages (TAM) expressed high levels of *PLAU* and *PLAUR*, and tumor-associated endothelial cells expressed high levels of *PLAT*, *F2R* and *SERPINE1* (Figure 4 and Figure 5). Of note, cancer cells were the major contributor to *F3* expression in OSCC. Dendritic cells and T cells expressed little, if any, of the core actors of the coagulome (Figure 4 and Figure 5). 

We further exploited scRNA-seq data to analyze the intra-tumoral heterogeneity of the coagulome in cancer cells. Firstly, the violin plot representation shown in Figure 6 confirmed the high inter-tumor heterogeneity that was observed with TCGA data: out of the ten tumors analyzed with scRNA-seq, four tumors strongly expressed *F3* and six expressed high levels of *PLAU*. In each individual tumor, great heterogeneity in *F3* and *PLAU* mRNA expression was found among cancer cells (Figure 6). Importantly, this intra-tumor heterogeneity led to the coexistence of «opposite» transcriptional programs, i.e., procoagulant (high levels of *F3*) vs. profibrinolytic (high levels of *PLAU*) within individual tumors. We examined the extent to which *F3* and *PLAU* expression overlapped in the cancer cells of each tumor (Figure 7). Within each OSCC, some cancer cells expressed both *F3* and *PLAU*. However, there were also separate contingents of cancer cells that presented opposite transcriptional programs, i.e., only *F3* or only *PLAU* (Figure 7). Depending on the tumor, the extent of the overlap in *F3* and *PLAU* mRNA expression varied greatly, from 12 to 84% of cancer cells. We next performed a comparison of the transcriptomic profiles of cancer cells exclusively expressing *F3* vs. *PLAU* with a GSEA analysis (Appendix A). A representative analysis, shown for example for one tumor (#20) (selected due to the availability of the highest number of cells), indicated that the intra-tumor heterogeneity was related to the hallmark «Epithelial-Mesenchymal Transition» (Appendix A). The data supported the existence of specific biological properties of cancer cells underpinning this striking intra-tumor heterogeneity and the coexistence of procoagulant and profibrinolytic programs. 

### 3.4. A Link between the Coagulome and the TME of OSCC

The coagulation cascade is a potent source of inflammation in the context of acute wound healing, and it may also have a role as a regulator of the TME [24,26]. We explored the possibility that tumors from TCGA, stratified according to their expression of *F3*, might differ in their overall cellular composition. We used the ESTIMATE algorithm to compare the content of stromal and immune cells in OSCC (Figure 8A). We found no significant differences in the stromal or immune scores between tumors with high vs. low *F3* expression (Figure 8A). These negative conclusions suggested that the coagulation process does not strongly alter the cellular composition of OSCC.

Next, we wanted to address the expression of the important immune checkpoints that control the tumor immune response and that are targeted with ICI [22]. Interestingly, an analysis using bulk RNA-seq data showed that OSCC with high *F3* (i.e., above median) expressed higher levels of the immune checkpoints *CD274*/PD-L1, *PDCD1LG2*/PD-L2 and *CD80*/B7-H1 (Figure 8B). We noted a 1.60-fold (*p* = 3.26 × 10^−4^), 1.86-fold (*p* = 4.43 × 10^−6^) and 1.42-fold (*p* = 2.04 × 10^−4^) increase in the median expression in high *F3*-expressing tumors, respectively. In order to validate and extend these conclusions obtained with bulk RNA-seq data from TCGA, we stratified the OSCC tumors analyzed with scRNA-seq into two groups according to *F3* expression in the cancer lineage (Figure 9). The violin plots shown in Figure 9 show the mRNA levels of the different checkpoints in the main cell types analyzed by Puram et al. [28]. Strikingly, tumors with a strong expression of *F3* in cancer cells displayed high levels of the immune checkpoints *CD274*/PD-L1, *PDCD1LG2*/PD-L2 and *CD80*/B7-H1 in the tumor-infiltrating dendritic cells (DC) (median levels of 4.45 in high *F3* vs. 0.10 in low *F3*; 4.01 in high *F3* vs. 0 in low *F3* and 4.78 in high *F3* vs. 0 in low *F3*, respectively) (Figure 9).

### 3.5. The Coagulome of OSCC Cells In Vitro

To validate our findings in vitro, we compared the coagulomes of three OSCC cell lines (PECA/PJ34, PECA/PJ41 and SCC9) to glioma cells (A172, SW1088 and U118MG) by immunoblotting (Figure 10A). High expression levels of TF and uPA were detected in the three OSCC cell lines, while tPA was only detected at faint levels. The data from genomic analyses provide a snapshot of basal tumor gene expression. We wondered whether the coagulome of OSCC might encounter variations upon exposure of OSCC to genotoxic therapeutic agents, such as the commonly used chemotherapeutics cisplatin, 5-fluorouracile and paclitaxel. We exposed OSCC cells to these chemotherapeutic agents applied at their previously determined IC_50_ concentrations [31]. We found no significant effect of these chemotherapeutic agents on TF/uPA expression, suggesting a relative stability of the coagulome in this context (Appendix A).

Next, we envisioned the possibility that the interaction of thrombin with its receptor PAR-1 might modulate the growth of OSCC cells in vitro. No expression of the PAR1 receptor was detected in any of the three OSCC cell lines. To examine the possibility that a small pool of PAR1 receptors expressed by OSCC cells might be functionally active, we exposed OSCC cells to thrombin (10 NIHU/L) and/or dabigatran (1 µM). We found no effect of thrombin on the growth of OSCC cells in this setting (Figure 10B). Neither the expression of the immune checkpoint CD274/PD-L1 nor the phosphorylation levels of the oncogenic kinases ERK1/2 were modulated by thrombin (Figure 10C). These results suggested that OSCC cells do not directly respond to thrombin. 

## 4. Discussion

Using genomic data from different sources, we examined the expression of the six genes that define the «core» coagulome, i.e., genes that contribute to coagulation and fibrinolysis in multiple, if not all, experimental systems [13]. We found surprisingly high expression levels of *F3* (TF) and *PLAU* (uPA), the two key antipodal actors of coagulation/fibrinolysis, in OSCC. Among the tumor types included in TCGA, OSCC was identified as the tumor type with the highest mRNA expression levels of *F3* and *PLAU*. Our in vitro studies confirmed the particularity of the coagulome of OSCC cells, further extending the conclusions of previous studies reporting that both TF and uPA are detectable at high protein levels in OSCC [13,32]. Overall, the coagulome of OSCC appears to be genetically stable. An inverse correlation between locus-specific DNA methylation and *F3*/*PLAU* mRNA was observed, in agreement with previous reports that documented the regulation of *F3* expression by DNA methylation in tumors other than OSCC [33]. Strikingly, we obtained data supporting the existence of great inter- and intra-tumor heterogeneity. At the single-cell level, we demonstrated that cancer cells with distinct pro-coagulant and pro-fibrinolytic transcriptional programs coexisted within each OSCC. We also found a potential link between the coagulation process and the TME of OSCC. The tumors with high expression levels of *F3* had higher levels of the key immune checkpoint molecules *CD274*/PD-L1, *PDCD1LG2*/PD-L2 and *CD80*. At the single-cell resolution, the increased expression of these immune checkpoint molecules was found to be striking in DC. 

Outside of the therapeutic context, OSCC is considered to be at low risk of systemic thrombotic events [7,8], a fact that has been discussed in recent literature as a paradox, considering its high expression levels of TF [8]. In the present study, we identified the high expression of *PLAU* (uPA), possibly counteracting clot formation via fibrinolysis, as an important element in the explanation of this paradox. Great inter-tumor heterogeneity was also observed. Our observations reinforce the notion that coagulation and fibrinolysis operate simultaneously in OSCC, in agreement with the observation that OSCC patients often have high preoperative levels of D-dimers [34]. Predicting which patients might benefit from anticoagulation in the therapeutic context remains challenging [1,2]. The finding that subpopulations of pro-coagulant and pro-fibrinolytic cancer cells coexist within individual OSCC illustrates the complex challenges posed by intra-tumor heterogeneity [35]. Based on our observations, we suggest that even OSCC with the highest global pro-coagulant transcriptomic profile might be at risk of local hemorrhage due to the presence of a fibrinolytic contingent. Our observations and those of others open up the possibility that EMT might be linked, or even contribute, to this heterogeneity [36]. More studies are required to explore the biological events that underpin the local and potentially dynamic regulation of the coagulome of OSCC. 

Besides the regulation of hemostasis, the coagulation cascade potentially holds broad implications for cancer biology. Coagulation may modulate cancer growth directly or target cells of the TME [15,19]. The existence of a direct regulation of cancer cells via PAR1 activation by thrombin was suggested in various types of solid tumors [37,38,39]. Its broad relevance is, however, a matter of discussion [40]. We found no expression of PAR1 (*F2R*) at the protein level in OSCC cells, and in our experimental conditions, thrombin had no detectable effect on the growth of OSCC cells, their activation of the oncogenic kinases ERK1/2 or the expression of the checkpoint molecule *CD274*/PD-L1. Meanwhile, our analysis of single-cell RNA-seq data indicated that *F2R*/PAR1 is strongly expressed in endothelial cells and cancer-associated fibroblasts, but not in cancer cells in OSCC. These data suggest that coagulation and thrombin, rather than directly modulating the behavior of OSCC cells, might exert some of their effects on the TME of these tumors. Importantly, the notion that the coagulation process might influence the different cell populations that constitute the TME is mounting [41]. The possibility that the tumor coagulome and the TME mutually regulate each other deserves further study when considering the increasingly recognized role of the TME as a therapeutic target [20,21].

We examined the possible impact of coagulation on the immune cells within the TME of OSCC [20,21]. We noted no significant correlation between the expression of *F3* in OSCC and the density of the infiltrate of cells of the adaptive immune system, suggesting that coagulation does not contribute to an «immune-excluded» phenotype in OSCC [23]. Importantly, however, tumors expressing high levels of *F3* had significantly higher levels of the immune checkpoint molecules *CD274*/PD-L1, *PDCD1LG2*/PD-L2 and *CD80*. The interaction between PD1 and its ligands PD-L1 and PD-L2 is a key checkpoint of the adaptive immune response against tumors [22]. Its targeting with nivolumab or pembrolizumab is approved for the treatment of recurrent/metastatic stages of HNSCC, as based on evidence obtained in phase 3 trials [22]. When we examined the tumor composition at the single-cell level, we found that DC displayed marked differences in the expression of these checkpoint molecules depending on the *F3* expression in the cancer cells of the tumor. This observation is important because DC are widely recognized as key antigen-presenting cells in the TME and they support anti-cancer adaptive immunity by activating T cells in the TME [42,43]. In the context of sepsis, a previous study reported PAR1 signaling on the surface of DC as a step that can amplify inflammation [44]. We did not detect *F2R*/PAR1 mRNA in DC in the single-cell RNA-seq analysis of OSCC, questioning the relevance of a direct control of DC by thrombin and the coagulation cascade in the TME. Our study, therefore, opens up the possibility that the coagulation cascade may indirectly regulate the expression of key immune checkpoint molecules in DC, for example by favoring a hypoxic microenvironment. The possibility that a high expression of *F3*/activation of coagulation might contribute to immune evasion in OSCC deserves to be experimentally tested in future studies. Importantly, recent studies are pointing to specific mechanisms through which some of the actors of the coagulation cascade may regulate tumor-specific adaptive immunity and antigen presentation [45]. Actioning the coagulome based on this rationale might be a promising strategy to boost the efficacy of the immune system, especially in cancer patients receiving immune checkpoint blockers [46]. 

## 5. Conclusions

OSCC express high levels of F3/TF and PLAU/uPA and these tumors constitute an interesting model for the study of coagulation in cancer. The study of the coagulome is nevertheless complex, as illustrated by the existence of great inter- and intra-tumor heterogeneity. Importantly, our findings suggest that the biological effects of the coagulome extend beyond the regulation of hemostasis. We observed that OSCC with high levels of F3 express high levels of the key immune checkpoint molecules CD274/PD-L1, PDCD1LG2/PD-L2 and CD80 in DC. Studying how the process of coagulation could be linked to the regulation of the local immune microenvironment promises to bring new insights on the regulation of the TME, with rich perspectives in terms of biomarkers and therapeutics.

## Figures and Tables

**Figure 1 cancers-14-00460-f001:**
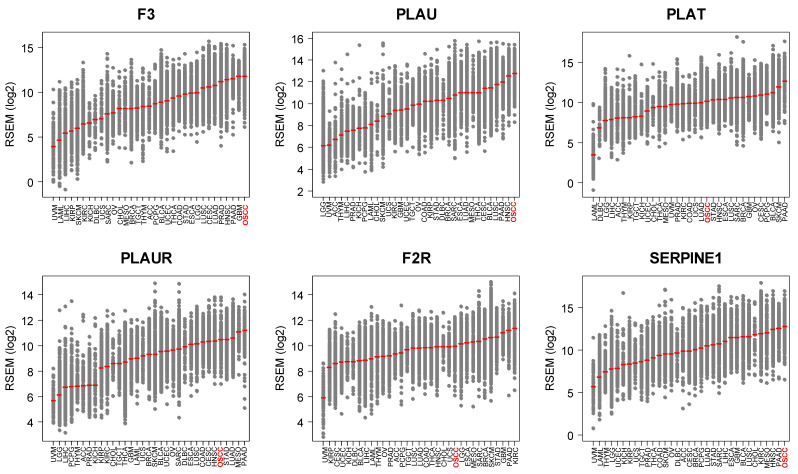
Coagulome gene expression in human tumors. Dot plots showing the tumor type ranking according to the mRNA expression levels of six essential components of the tumor coagulome (*F3*, *PLAU*, *PLAT*, *PLAUR*, *F2R* and *SERPINE1*). Data from *n* = 33 tumor types, including OSCC, were retrieved from TCGA (total number of tumors analyzed was *n* = 10,071). Note that OSCC is the tumor type that presents the highest expression levels of *F3*, *PLAU* and *SERPINE1* (OSCC in red).

**Figure 2 cancers-14-00460-f002:**
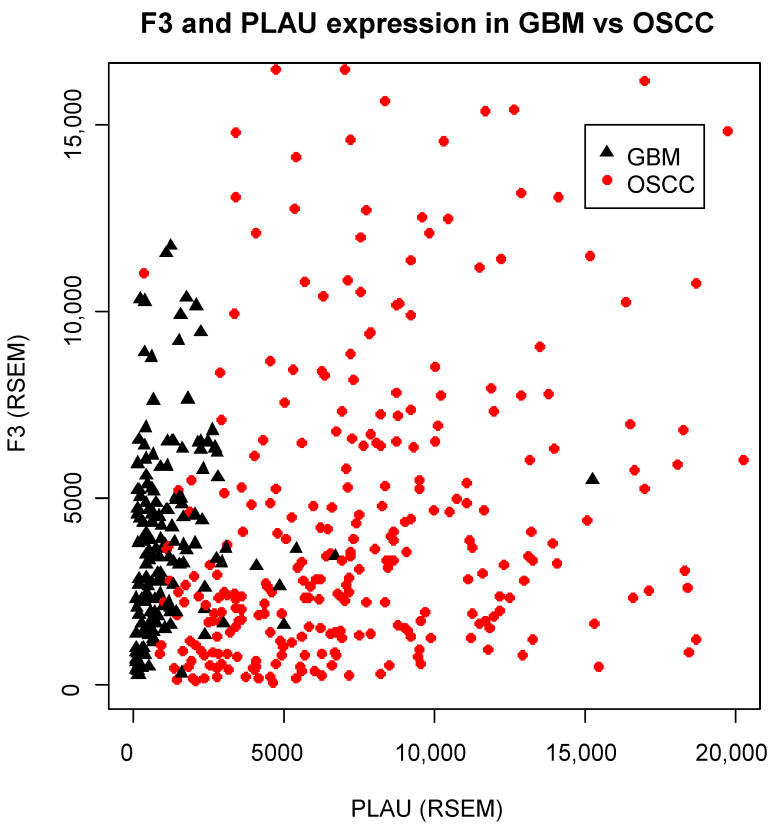
Inter-tumor heterogeneity of *F3* and *PLAU* expression in OSCC and GBM. Comparison of *F3* and *PLAU* expression in OSCC (red, *n* = 321) and GBM (black, *n* = 166) from TCGA. mRNA levels are shown as RSEM, showing the great inter-tumor heterogeneity of OSCC tumors. Note that each dot represents an individual tumor.

**Figure 3 cancers-14-00460-f003:**
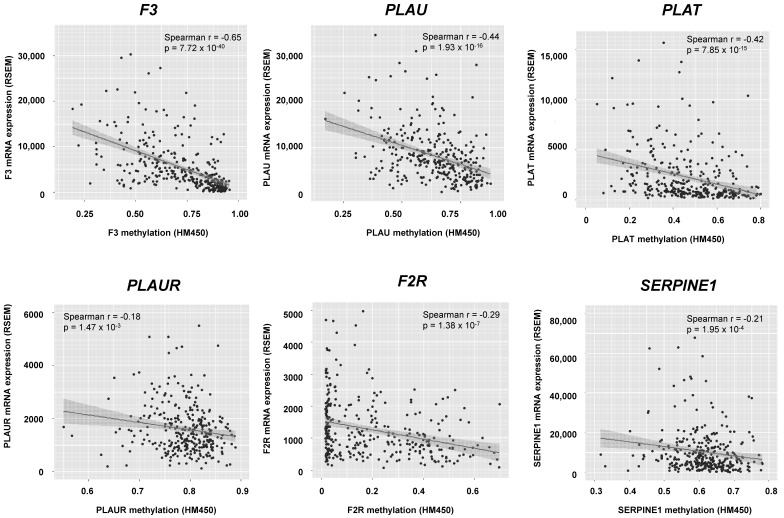
Link between coagulome gene expression and gene methylation. Local DNA methylation was negatively correlated with *F3*, *PLAU* and *PLAT* gene expression. Graphs showing *F3*, *PLAU* and *PLAT* mRNA expressions (RSEM) and the CpG methylation levels of the gene promoter (HM450 beta values, 0 = unmethylated, 1 = full methylated). Correlations were performed using Spearman analysis.

**Figure 4 cancers-14-00460-f004:**
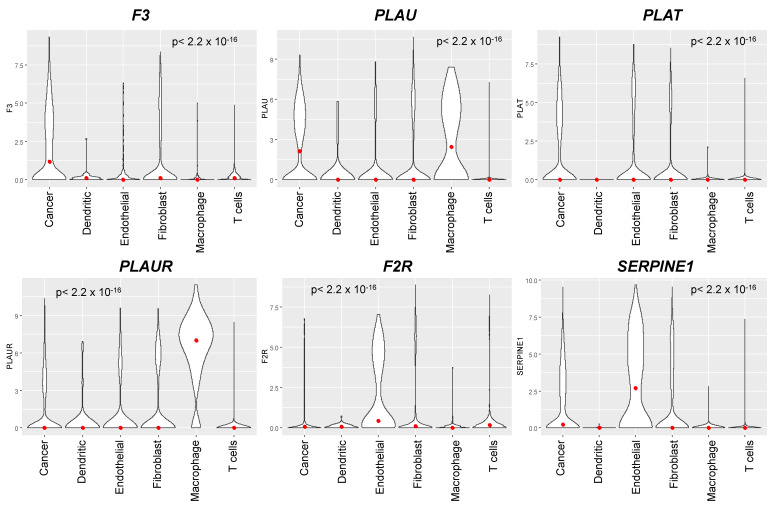
Single-cell analysis of the coagulome in different cell subpopulations in OSCC. Violin plots showing the mRNA expression of the core coagulome genes (*F3*, *PLAU*, *PLAT*, *PLAUR*, *F2R* and *SERPINE1*) in different cell populations present in OSCC (malignant cancer cells, T cells, dendritic cells, macrophages, endothelial cells and fibroblasts). Data are shown from OSCC tumors analyzed by scRNA-seq in Puram et al. [28] (GSE103322). The red dots represent the median values. The indicated *p* values were obtained with the Kruskal–Wallis test.

**Figure 5 cancers-14-00460-f005:**
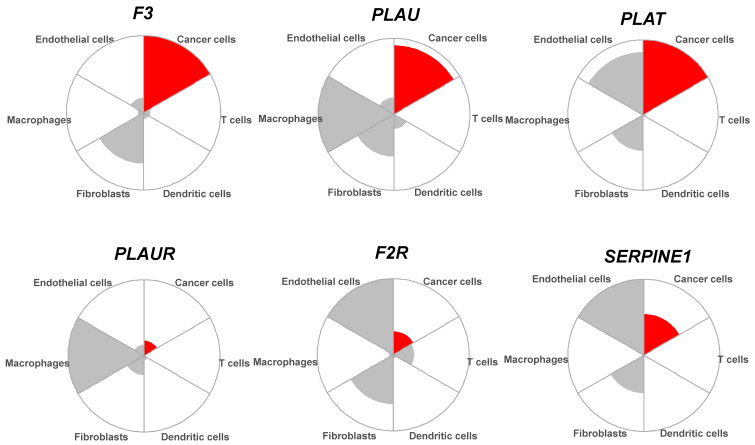
Polar plots showing the mRNA expression of the core coagulome genes (*F3*, *PLAU*, *PLAT*, *PLAUR*, *F2R* and *SERPINE1*) in different cell subpopulations of OSCC. The graphs show the relative expression of each gene in the different cell types (malignant cancer cells, T cells, dendritic cells, macrophages, endothelial cells and fibroblasts) using scRNA-seq data from GSE103322. In each case, the cell type with the highest expression (median) was set as maximum. Note that cancer cells are shown in red.

**Figure 6 cancers-14-00460-f006:**
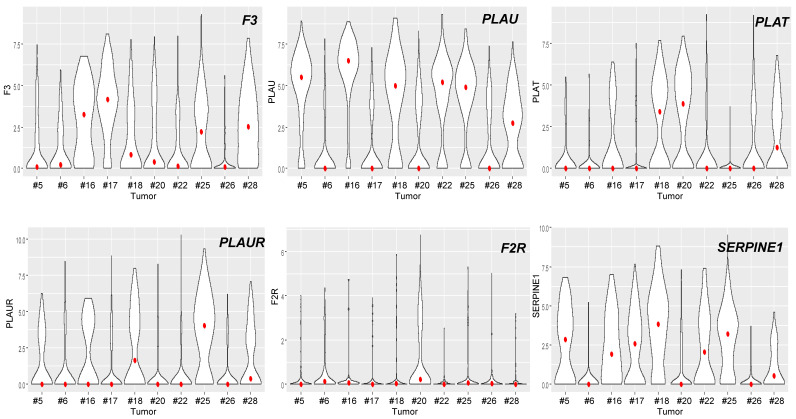
Intra-tumor heterogeneity of the coagulome of malignant cells from individual OSCC. Violin plots showing the expression of the core coagulome genes (*F3*, *PLAU*, *PLAT*, *PLAUR*, *F2R* and *SERPINE1*) in cancer cells from different OSCC tumors analyzed in GSE103322. Each individual tumor was identified according to the study by Puram et al. [28]. The red dots represent the median value for the expression of each gene.

**Figure 7 cancers-14-00460-f007:**
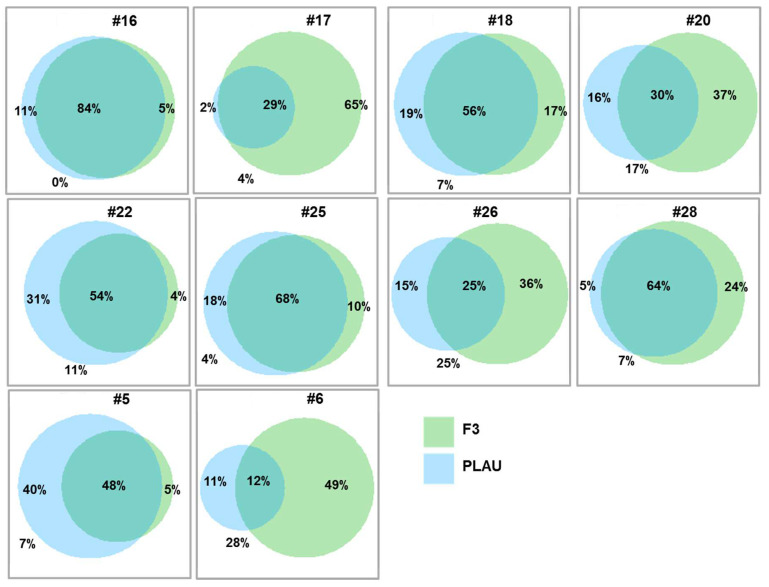
Overlap in *F3* and *PLAU* expression in cancer cells from individual tumors. Venn diagrams examining the overlap between the pro-coagulant (*F3*-positive) and pro-fibrinolytic (*PLAU*-positive) transcriptional programs in cancer cells from individual tumors (GSE103322). Note that the extent of the overlap of *F3* and *PLAU* expression varied greatly depending on the tumor analyzed, ranging from 12% to 84% of cancer cells.

**Figure 8 cancers-14-00460-f008:**
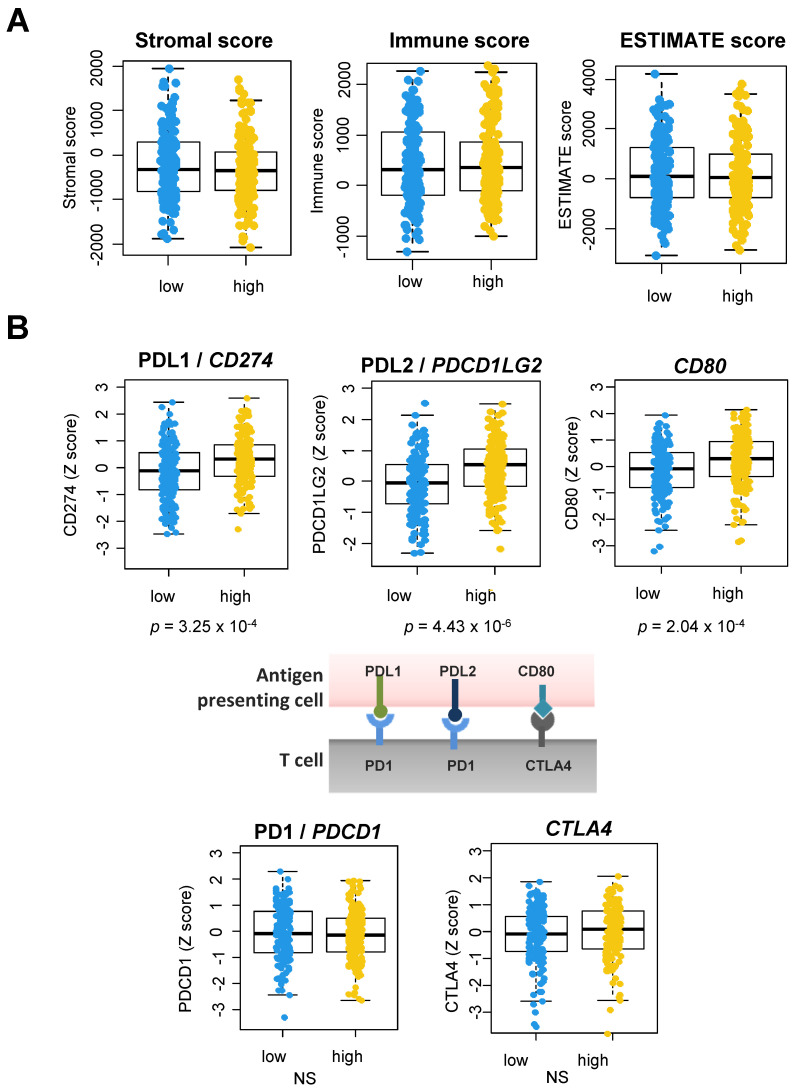
Stromal and immune infiltration and immune checkpoint expression in OSCC tumors. (**A**) Using TCGA data, the stromal and immune infiltrate was assessed in low *F3* (<median, *n* = 158) vs. high *F3* (>median, *n* = 158) OSCC tumors from TCGA using the ESTIMATE algorithm. (**B**) Boxplots showing the expression of immune checkpoint molecules (*CD274*/PD-L1, *PDCD1LG2*/PD-L2, *CD80*, *PDCD1*/PD1, *CTLA4*) in OSCC with low *F3* vs. high *F3* expression.

**Figure 9 cancers-14-00460-f009:**
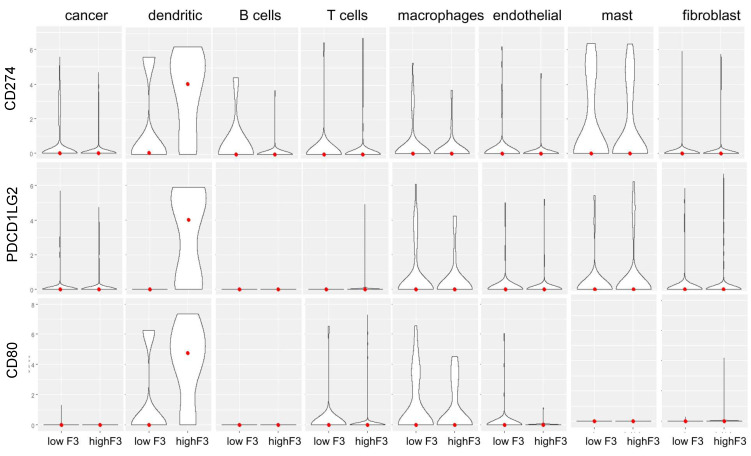
Single-cell analysis of the expression of immune checkpoint molecules in tumors stratified according to the expression of *F3* in cancer cells. Violin plots showing the expression of *CD274*/PD-L1, *PDCD1LG2*/PD-L2 and *CD80* in the different cell populations from the GSE103322 cohort, including cancer cells, B cells, T cells, macrophages, endothelial cells, fibroblasts, dendritic cells, mast cells.

**Figure 10 cancers-14-00460-f010:**
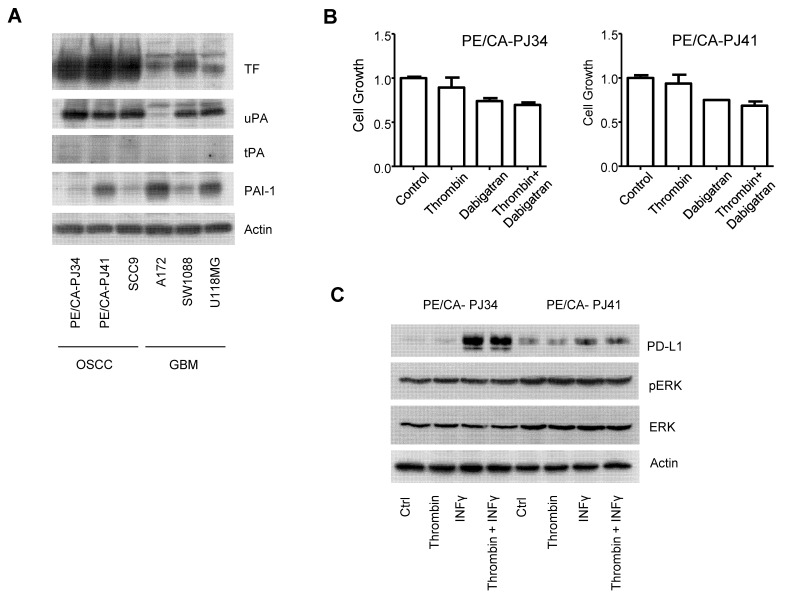
Immunoblot analysis of the coagulome in OSCC and the lack of a direct effect of thrombin on the growth of OSCC cells. (**A**) Immunoblot analysis of the expression of the Tissue Factor (*F3*), uPA (*PLAU*), tPA (*PLAT*) and PAI-1 (*SERPINE1*) in OSCC cell lines (PE/CA-PJ34, PE/CA-PJ41, SCC9) compared to GBM cell lines (A172, SW1088, U118MG). Actin was provided as a control. Protein extracts were prepared from the indicated cell lines as indicated. Note that the blots are representative of at least 3 independent experiments. (**B**) Lack of a direct effect of thrombin on the growth of OSCC cells in vitro. Thrombin (10 NIHU/L) and dabigatran (1 µM) were maintained for 72 h, and cancer cell growth was measured with crystal violet and normalized by taking the control condition as reference. (**C**) Lack of a direct effect of thrombin on the expression of the immune checkpoint molecule CD274/PD-L1 and ERK phosphorylation in OSCC cells. Thrombin (10 NIHU/L) was applied at the same time as Interferon-Gamma (10 ng/mL), used here as an inducer of PD-L1 expression. The indicated markers were analyzed after 18 h of incubation.

## Data Availability

The data from TCGA and GSE103322 are publicly available. The datasets used in the current study are available from the corresponding author.

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
