# Peer review of "Molecular Landscape of the Coagulome of Oral Squamous Cell Carcinoma"

_cancers, 2022, doi:10.3390/cancers14020460_

Round 1

Reviewer 1 Report

The article gives an overall description of the OSCC (epi)genetic characteristics in terms of coagulome components. Well-written and structured, the article highlight how heterogenous can be the expression of coagulome components inter-tumor, as well as within the cells of the TME. For these reasons, the majority of my comments are mostly direct on improving few figures, already of excellent quality, and highlight few concepts, which can be extended at the end of each paragraph or in the discussion.

Nevertheless, I consider the last paragraph as the weak point of this study, where the descriptive part meets the experimental one (see specific comments).

Results

Suppl. Figure 1. The author should add the extended name of GBM, before the abbreviation.

Suppl. Figure 2. Exchange “Grade” with “G”, as done for tumor (T) and nodal (N).

Suppl. Figure 3. So far the two parameters, smoke and age, have been analyzed separately, a combined analysis might reveal a specific coagulome fingerprint.

Figure 3. The graph is of visual effect, but gives little information about the degree of differences in gene expression between the different cell types. Furthermore, the expression of different genes when focusing on a specific cell type cannot be compared. I would suggest to create a hierarchical clustering heatmap to improve the visual representation of the results. Furthermore, depending on the original data source, create a violin plot where statistical significance of the expression of a gene within between different cell types can be highlighted.

Figure 4A. “Violin plots showing the mRNA expression of the core coagulome genes (F3, PLAU, PLAT, PLAUR, F2R and SERPINE1)”, only F3 and PLAU are showed.

The names of samples are not aligned with the bars of the violin plots.

“…five tumor strongly express F3”, I would say four. Sample #18 cannot be considered as strongly express F3, although above 0. Furthermore, the other four express F3 at a much higher levels, cannot be compared or included in the same category of #18.

Figure 4B. Representation of F3 and PLAU intra- and inter-tumor expression. I have a question about the sample selection. Based on Figure 4A., samples #17 has high F3 and low PLAU, #26 both low and #20 both low as well, would be better to have one with low F3 and high PLAU, like #22?

Figure 4C. Why is, in general, samples #20 taken in consideration? It looks the “worst” in terms of median expression of both F3 and PLAU, they are both low.

Figure 6. Is it known what kind of DC are typically found in OSCC microenvironment?

The coagulome of OSCC cells in vitro.

Figure 7A. I would suggest grouping the different cell type under OSCC and GBM by writing it underneath.

Figure 7B. Name of the figure in the text is wrong. The material and methods of this part need to be improved, how long was the treatment? which concentration? The results are based on protein levels, while all the previous descriptive analysis where based on gene expression (RNAseq), which is much more sensitive. The author might lose important changes on the gene expression profile. Based on Suppl. Figure 4., F3, PLAU and PLAT gene expression does not affect OS or DFS. In line with the introduction, the coagulome has an impact during the treatment, when the dynamic between coagulation and fibrinolysis factors is disrupted. In fact, as mentioned in this study, the coagulome of OSCC appears to be normally stable. Thus, this is the moments in which the author should really show that, after cellular disruption (in this case chemotherapeutic drugs have been chosen), there is a considerable and important alteration of the coagulome components. Otherwise, what is the importance of this profile in terms of therapeutic strategy? It is essential to establish in which circumstances the coagulome dynamic is altered. Also, based on Figure 3, the coagulome gene expression is spread both on tumor and microenvironment (ME) cells. Furthermore, DCs and not the tumor cells represent the major immune suppressive component in the OSCC ME. Thus, if really nothing change in the coagulome of OSCC cells upon treatment, the author should show that coagulome variations occurs in ME cells (e.g. fibroblast and macrophages) and ICP alterations in DCs.

Altogether, this paper offer a broad descriptive analysis of OSCC coagulome gene expression based on TCGA and previous publish data. At this point, it is acceptable and in line with the title, offering an overall idea of the oral squamous cell carcinoma coagulome. Nevertheless, the last paragraph appears much weaker than the others, making drop the overall impact and quality of the paper.

Author Response

1/ The article gives an overall description of the OSCC (epi)genetic characteristics in terms of coagulome components. Well-written and structured, the article highlight how heterogenous can be the expression of coagulome components inter-tumor, as well as within the cells of the TME. For these reasons, the majority of my comments are mostly direct on improving few figures, already of excellent quality, and highlight few concepts, which can be extended at the end of each paragraph or in the discussion.

We thank the reviewer for these positive comments and the time spent on reading / commenting on our study. We are now proposing a revised version that addresses the precise comments and suggestions made by the reviewer.

2/ Nevertheless, I consider the last paragraph as the weak point of this study, where the descriptive part meets the experimental one (see specific comments).

We understand the reviewer’s comment. The last paragraph, describing the experimental conclusions obtained in vitro, does not reinforce all the conclusions made in silico, especially those regarding the TME of OSCC.

Nevertheless, we respectfully ask the permission to maintain this set of experiments. Indeed, they provide a validation for one important finding of the study, i.e. the expression of high levels of TF and uPA in OSCC cells. They also extend our conclusions, by extrapolating our findings from functional genomics to the protein level. Last but not least, we are not aware of any previous studies examining the response of OSCC cells to thombin / Direct Oral Anticoagulants (DOAC). We strongly believe that these results, that are negative but conclusive, will be useful to the OSCC research community.

Finally, we have also simplified the presentation and clarified the aims of the in vitro study (see p. 16 of the revised manuscript). Some data were moved to supplementary material (Suppl. Fig. 5).

3/ Results : Suppl. Figure 1. The author should add the extended name of GBM, before the abbreviation.

We agree and apologize for not having introduced the abbreviation for GBM in the initial version of the manuscript. This has been corrected in the revised version of the manuscript (p. 12).

4/ Suppl. Figure 2. Exchange “Grade” with “G”, as done for tumor (T) and nodal (N).

This is now done as indicated. Please note that the former Suppl. Fig. 2 is now the Suppl. Fig. 1 in the revised manuscript.

5/ Suppl. Figure 3. So far the two parameters, smoke and age, have been analyzed separately, a combined analysis might reveal a specific coagulome fingerprint.

We have performed a new set of analyses to address the reviewer’s request. We have however obtained no hint regarding the interest of combining smoking and age in the analysis (data not shown). If the reviewer insists, we could insert the corresponding data, either as a statement « data not shown » or as supplementary material.

6/ Figure 3. The graph is of visual effect, but gives little information about the degree of differences in gene expression between the different cell types. Furthermore, the expression of different genes when focusing on a specific cell type cannot be compared. I would suggest to create a hierarchical clustering heatmap to improve the visual representation of the results. Furthermore, depending on the original data source, create a violin plot where statistical significance of the expression of a gene within between different cell types can be highlighted.

We do agree with the reviewer that a violin plot presentation brings the possibility to perform statistical analyses. The requested violin plots can be found in the new figure 4. The existence of statistical differences between cell types is mentioned here. We are grateful to the reviewer for this suggestion and the new presentation that strengthens our conclusions.

Following the reviewer’s request, we have also created a heatmap to present the coagulome of the different cell types present within OSCC (shown below).

While a heatmap is useful to analyse large data sets and to make sense of complex variations, we think it is not really interesting here. The heatmap does not bring information beyond  the description of the cell-specific pattern of expression of the coagulome in OSCC, i.e. information already provided by the polar plots. Because the polar plots provide a nicer visualisation than the heatmap, we have not inserted the corresponding panel in the revised manuscript. We are nevertheless ready to insert it if the reviewer insists.

7/ Figure 4A. “Violin plots showing the mRNA expression of the core coagulome genes (F3, PLAU, PLAT, PLAUR, F2R and SERPINE1)”, only F3 and PLAU are showed.

The names of samples are not aligned with the bars of the violin plots.

“…five tumor strongly express F3”, I would say four. Sample #18 cannot be considered as strongly express F3, although above 0. Furthermore, the other four express F3 at a much higher levels, cannot be compared or included in the same category of #18.

Thank you for having noticed these mistakes. We have inserted the requested plots as the new Figure 6. Thank you also for raising our attention to the design of the previous versions of the figures, where the names of the samples were indeed not aligned. This has been corrected as indicated.

We agree with the reviewer’s comment regarding the interpretation of F3 expression. We have corrected the manuscript to indicate that four out of ten tumors express high levels of F3 (page 14 of the revised manuscript). 

8/ Figure 4B. Representation of F3 and PLAU intra- and inter-tumor expression. I have a question about the sample selection. Based on Figure 4A., samples #17 has high F3 and low PLAU, #26 both low and #20 both low as well, would be better to have one with low F3 and high PLAU, like #22?

We apologize for poorly explaining our point here. Our initial choice of the three patients was guided by the number of single cells available. The three patients that were shown in Fig. 4B were those with the highest number of cells. 

In the revised version of the manuscript, we have changed the presentation of the figures to give a direct and global representation of the data, showing all tumors, without selecting any specific subset of tumors. The former panels 4B and 4C are replaced by a new figure 7 that shows the results for all tumors. We are grateful for this suggestion, as this will indeed prevent the readers from questioning the existence of a selection bias. 

9/ Figure 4C. Why is, in general, samples #20 taken in consideration? It looks the “worst” in terms of median expression of both F3 and PLAU, they are both low.

Once again, sample #20 was selected because of the large number of cancer cells that were available. The former panels 4B and 4C are replaced by the new figure 7 that shows the results for all tumors.

10/ Figure 6. Is it known what kind of DC are typically found in OSCC microenvironment?

We thank the reviewer for his/her interest in our findings regarding DC. DC are a cell type with a key function in adaptive immunity, but they are indeed heterogeneous. In OSCC, one subset of DC known as plasmacytoid DC is commonly found within the TME. This subset of DC is known to be tolerogenic. This phenotype is however under complex regulation, and it is unclear to which extent it is a stable cellular property, or one related to the local conditions found in the TME.

In our study, we refrained from making conclusions regarding the different subsets of DCs, since we could only use transcriptomic data. While the point raised by reviewer #1 is interesting, we propose to leave further investigation on the regulation of subpopulations of DC by coagulation for future studies.

11/ The coagulome of OSCC cells in vitro. Figure 7A. I would suggest grouping the different cell type under OSCC and GBM by writing it underneath.

Figure 7B. Name of the figure in the text is wrong. The material and methods of this part need to be improved, how long was the treatment? which concentration? The results are based on protein levels, while all the previous descriptive analysis where based on gene expression (RNAseq), which is much more sensitive. The author might lose important changes on the gene expression profile.

We have modified the labelling of the former Fig. 7A (now Fig. 10) as requested. We agree that this modification facilitates its interpretation.

The former panel B of Fig. 7 is now presented as a suppl. Fig. 5. We have expanded the figure legend, and we now include the concentration of each chemotherapeutic agent used as well as the exposure time (48h). Thank you for this excellent comment. 

Regarding the usefulness of protein analysis, we agree with the reviewer that protein analysis is not the most sensitive or practical tool to study the regulation of TF/uPA. Nevertheless, we think that a validation of our genomic conclusions at the protein level is important. The observation that high levels of F3/PLAU mRNA convert to high expression of TF/uPA in OSCC validates the use of genomics, as reported in the manuscript.

12/ Based on Suppl. Figure 4., F3, PLAU and PLAT gene expression does not affect OS or DFS. In line with the introduction, the coagulome has an impact during the treatment, when the dynamic between coagulation and fibrinolysis factors is disrupted. In fact, as mentioned in this study, the coagulome of OSCC appears to be normally stable. Thus, this is the moments in which the author should really show that, after cellular disruption (in this case chemotherapeutic drugs have been chosen), there is a considerable and important alteration of the coagulome components. Otherwise, what is the importance of this profile in terms of therapeutic strategy? It is essential to establish in which circumstances the coagulome dynamic is altered.

We are grateful for the interesting discussion raised by the reviewer. Reading the comments of the reviewer, we realize that our use of the term « dynamic » might have been inappropriate. We meant to underline that coagulation and fibrinolysis coexist, and that any blood clots would rapidly get dissolved. We did not mean that the coagulome of individual tumors is labile. In fact, we have not observed any strong experimental modulation of the coagulome in OSCC in vitro. In order to prevent this misinterpretation, we have removed the use of the term « dynamic coagulome ». The data on the coagulome in OSCC cells were also moved to the suppl. section, as a new suppl. Fig. 5. We think that, by so doing, the manuscript is more focused. We are grateful to the reviewer for helping us to clarify the presentation of our data.

13/ Also, based on Figure 3, the coagulome gene expression is spread both on tumor and microenvironment (ME) cells. Furthermore, DCs and not the tumor cells represent the major immune suppressive component in the OSCC ME. Thus, if really nothing change in the coagulome of OSCC cells upon treatment, the author should show that coagulome variations occurs in ME cells (e.g. fibroblast and macrophages) and ICP alterations in DCs.

This is again an important point. Indeed, we are reporting an analysis of the coagulome of the main cell types that constitute the TME. We anticipate that any change in tumor composition that would be induced for example by drug treatment would shape the coagulome in the long term. Importantly however, such changes cannot be easily studied in vitro. Once again, in the revised manuscript, we have rephrased the discussion to avoid any speculation on the stability of the coagulome. Clearly, more studies are required to address this difficult question.

14/ Altogether, this paper offer a broad descriptive analysis of OSCC coagulome gene expression based on TCGA and previous publish data. At this point, it is acceptable and in line with the title, offering an overall idea of the oral squamous cell carcinoma coagulome. Nevertheless, the last paragraph appears much weaker than the others, making drop the overall impact and quality of the paper.

Once again, we would like to thank the reviewer for carefully reading and commenting our manuscript. We do appreciate the useful comments and suggestions that we have received. Our reply regarding the last paragraph of the study can be found in detail in points #2 and #11 of this letter. We have worked on the last paragraph to improve its link to the rest of the study.

Reviewer 2 Report

The article “Molecular landscape of the coagulome of oral squamous cell carcinoma” by Lottin et al. deals with a very interesting topic. It explores the regulation of activators and inhibitors of coagulation in the various cell types in OSCC. Overall, the study is well written, the aims are clearly stated, and the results are depicted adequately. However, we must be aware that few data were collected by the authors themselves. The study follows almost completely a data mining approach. The sources of the analysed data sets are named precisely, however, and such an approach can of course also provide valuable insights. Nevertheless, I am missing a bit of what can be concluded from the findings and what significance the found features and correlations have or what clinical, therapeutic, or even basic scientific conclusions can now be drawn.

The introduction is well written, and it succeeds in introducing the reader to the topic.

The materials and methods section clearly states the used methods, however, for readers that are not that familiar with the used approaches it could be more detailed. As MDPI Cancers is no journal that only deals with RNA-seq data and transcriptomics from my point of view, two or three explanatory sentences on the methods and what they can do would be useful for the general readership.

The results section is well written and overall the results are depicted clearly. However, I wondered why many interesting results are hidden in the supplementary files.
To make it concrete the following questions came up for me:
- Why is the Suppl. Fig. 1 not included in the main text?
- In Suppl. Fig. 1 what does GBM stand for? Glioblastoma? As far as I see you did not introduce this abbreviation.
- For readers that are not familiar with that kind of presentation, the caption of Fig. 3 should explain in more detail what is depicted here.
- “… We examined the extent to which F3 and PLAU expression overlapped in each tumor (Suppl. Figure 5)…” – do you mean that the same cells express high F3 and high PLAU? This should be described more clearly.
- In Fig. 4A – what do the two most right-handed elements of the diagram represent where no tumor number is given? As far as I see, this is described nowhere.
- Fig. 4B is too small in my opinion. I would suggest, to show instead the complete Suppl. Fig. 5 because for me it was easier to understand the findings with that figure.
- In Figure 5A it was not clear to me what the panel with the ESTIMATE score shows.
- You used three chemotherapeutic drugs in order to analyse whether they influence TF or uPA. Why? What was the rationale behind this experiment?
- I miss the n/N in Fig. 7, are these exemplary data?
- I have several questions and comments on the blots in the supplementary material. I miss any labelling. You can only guess what you are supposed to see. In Panel A how do you explain all these additional bands? Why are not all lanes shown in the manuscript? The blot detecting TF in PE/CA PJ34 seems to me as if it is not the blot shown in the manuscript, why? The blots corresponding to PE/CA PJ41 appear to be slightly compressed or stretched as far as I can see. Please double-check all blots that everything is correct and please add a clear labelling to the blots also including all probes and the molecular weight marker.

The discussion is well written, and it contains many valuable cross-links to previous studies. As already described, it is not entirely clear to me in the manuscript what valuable conclusions can be drawn from the findings. It is often said that the tumors have a high variability, which unfortunately is not a new finding for OSCC. For me, clearer conclusions would be needed from the correlations found or at least a concrete outlook as to what is to be investigated in the future and for what purpose.
- “Thrombotic accidents” should be “thrombotic incidents” shouldn’t it? 

Author Response

1/ The article gives an overall description of the OSCC (epi)genetic characteristics in terms of coagulome components. Well-written and structured, the article highlight how heterogenous can be the expression of coagulome components inter-tumor, as well as within the cells of the TME. For these reasons, the majority of my comments are mostly direct on improving few figures, already of excellent quality, and highlight few concepts, which can be extended at the end of each paragraph or in the discussion.

We thank the reviewer for these positive comments and the time spent on reading / commenting on our study. We are now proposing a revised version that addresses the precise comments and suggestions made by the reviewer.p

2/ Nevertheless, I consider the last paragraph as the weak point of this study, where the descriptive part meets the experimental one (see specific comments).

We understand the reviewer’s comment. The last paragraph, describing the experimental conclusions obtained in vitro, does not reinforce all the conclusions made in silico, especially those regarding the TME of OSCC.

Nevertheless, we respectfully ask the permission to maintain this set of experiments. Indeed, they provide a validation for one important finding of the study, i.e. the expression of high levels of TF and uPA in OSCC cells. They also extend our conclusions, by extrapolating our findings from functional genomics to the protein level. Last but not least, we are not aware of any previous studies examining the response of OSCC cells to thombin / Direct Oral Anticoagulants (DOAC). We strongly believe that these results, that are negative but conclusive, will be useful to the OSCC research community.

Finally, we have also simplified the presentation and clarified the aims of the in vitro study (see p. 16 of the revised manuscript). Some data were moved to supplementary material (Suppl. Fig. 5).

3/ Results : Suppl. Figure 1. The author should add the extended name of GBM, before the abbreviation.

We agree and apologize for not having introduced the abbreviation for GBM in the initial version of the manuscript. This has been corrected in the revised version of the manuscript (p. 12).

4/ Suppl. Figure 2. Exchange “Grade” with “G”, as done for tumor (T) and nodal (N).

This is now done as indicated. Please note that the former Suppl. Fig. 2 is now the Suppl. Fig. 1 in the revised manuscript.

5/ Suppl. Figure 3. So far the two parameters, smoke and age, have been analyzed separately, a combined analysis might reveal a specific coagulome fingerprint.

We have performed a new set of analyses to address the reviewer’s request. We have however obtained no hint regarding the interest of combining smoking and age in the analysis (data not shown). If the reviewer insists, we could insert the corresponding data, either as a statement « data not shown » or as supplementary material.

6/ Figure 3. The graph is of visual effect, but gives little information about the degree of differences in gene expression between the different cell types. Furthermore, the expression of different genes when focusing on a specific cell type cannot be compared. I would suggest to create a hierarchical clustering heatmap to improve the visual representation of the results. Furthermore, depending on the original data source, create a violin plot where statistical significance of the expression of a gene within between different cell types can be highlighted.

We do agree with the reviewer that a violin plot presentation brings the possibility to perform statistical analyses. The requested violin plots can be found in the new figure 4. The existence of statistical differences between cell types is mentioned here. We are grateful to the reviewer for this suggestion and the new presentation that strengthens our conclusions.

Following the reviewer’s request, we have also created a heatmap to present the coagulome of the different cell types present within OSCC (shown below).

While a heatmap is useful to analyse large data sets and to make sense of complex variations, we think it is not really interesting here. The heatmap does not bring information beyond  the description of the cell-specific pattern of expression of the coagulome in OSCC, i.e. information already provided by the polar plots. Because the polar plots provide a nicer visualisation than the heatmap, we have not inserted the corresponding panel in the revised manuscript. We are nevertheless ready to insert it if the reviewer insists.

7/ Figure 4A. “Violin plots showing the mRNA expression of the core coagulome genes (F3, PLAU, PLAT, PLAUR, F2R and SERPINE1)”, only F3 and PLAU are showed.

The names of samples are not aligned with the bars of the violin plots.

“…five tumor strongly express F3”, I would say four. Sample #18 cannot be considered as strongly express F3, although above 0. Furthermore, the other four express F3 at a much higher levels, cannot be compared or included in the same category of #18.

Thank you for having noticed these mistakes. We have inserted the requested plots as the new Figure 6. Thank you also for raising our attention to the design of the previous versions of the figures, where the names of the samples were indeed not aligned. This has been corrected as indicated.

We agree with the reviewer’s comment regarding the interpretation of F3 expression. We have corrected the manuscript to indicate that four out of ten tumors express high levels of F3 (page 14 of the revised manuscript). 

8/ Figure 4B. Representation of F3 and PLAU intra- and inter-tumor expression. I have a question about the sample selection. Based on Figure 4A., samples #17 has high F3 and low PLAU, #26 both low and #20 both low as well, would be better to have one with low F3 and high PLAU, like #22?

We apologize for poorly explaining our point here. Our initial choice of the three patients was guided by the number of single cells available. The three patients that were shown in Fig. 4B were those with the highest number of cells. 

In the revised version of the manuscript, we have changed the presentation of the figures to give a direct and global representation of the data, showing all tumors, without selecting any specific subset of tumors. The former panels 4B and 4C are replaced by a new figure 7 that shows the results for all tumors. We are grateful for this suggestion, as this will indeed prevent the readers from questioning the existence of a selection bias. 

9/ Figure 4C. Why is, in general, samples #20 taken in consideration? It looks the “worst” in terms of median expression of both F3 and PLAU, they are both low.

Once again, sample #20 was selected because of the large number of cancer cells that were available. The former panels 4B and 4C are replaced by the new figure 7 that shows the results for all tumors.

10/ Figure 6. Is it known what kind of DC are typically found in OSCC microenvironment?

We thank the reviewer for his/her interest in our findings regarding DC. DC are a cell type with a key function in adaptive immunity, but they are indeed heterogeneous. In OSCC, one subset of DC known as plasmacytoid DC is commonly found within the TME. This subset of DC is known to be tolerogenic. This phenotype is however under complex regulation, and it is unclear to which extent it is a stable cellular property, or one related to the local conditions found in the TME.

In our study, we refrained from making conclusions regarding the different subsets of DCs, since we could only use transcriptomic data. While the point raised by reviewer #1 is interesting, we propose to leave further investigation on the regulation of subpopulations of DC by coagulation for future studies.

11/ The coagulome of OSCC cells in vitro. Figure 7A. I would suggest grouping the different cell type under OSCC and GBM by writing it underneath.

Figure 7B. Name of the figure in the text is wrong. The material and methods of this part need to be improved, how long was the treatment? which concentration? The results are based on protein levels, while all the previous descriptive analysis where based on gene expression (RNAseq), which is much more sensitive. The author might lose important changes on the gene expression profile.

We have modified the labelling of the former Fig. 7A (now Fig. 10) as requested. We agree that this modification facilitates its interpretation.

The former panel B of Fig. 7 is now presented as a suppl. Fig. 5. We have expanded the figure legend, and we now include the concentration of each chemotherapeutic agent used as well as the exposure time (48h). Thank you for this excellent comment. 

Regarding the usefulness of protein analysis, we agree with the reviewer that protein analysis is not the most sensitive or practical tool to study the regulation of TF/uPA. Nevertheless, we think that a validation of our genomic conclusions at the protein level is important. The observation that high levels of F3/PLAU mRNA convert to high expression of TF/uPA in OSCC validates the use of genomics, as reported in the manuscript.

12/ Based on Suppl. Figure 4., F3, PLAU and PLAT gene expression does not affect OS or DFS. In line with the introduction, the coagulome has an impact during the treatment, when the dynamic between coagulation and fibrinolysis factors is disrupted. In fact, as mentioned in this study, the coagulome of OSCC appears to be normally stable. Thus, this is the moments in which the author should really show that, after cellular disruption (in this case chemotherapeutic drugs have been chosen), there is a considerable and important alteration of the coagulome components. Otherwise, what is the importance of this profile in terms of therapeutic strategy? It is essential to establish in which circumstances the coagulome dynamic is altered.

We are grateful for the interesting discussion raised by the reviewer. Reading the comments of the reviewer, we realize that our use of the term « dynamic » might have been inappropriate. We meant to underline that coagulation and fibrinolysis coexist, and that any blood clots would rapidly get dissolved. We did not mean that the coagulome of individual tumors is labile. In fact, we have not observed any strong experimental modulation of the coagulome in OSCC in vitro. In order to prevent this misinterpretation, we have removed the use of the term « dynamic coagulome ». The data on the coagulome in OSCC cells were also moved to the suppl. section, as a new suppl. Fig. 5. We think that, by so doing, the manuscript is more focused. We are grateful to the reviewer for helping us to clarify the presentation of our data.

13/ Also, based on Figure 3, the coagulome gene expression is spread both on tumor and microenvironment (ME) cells. Furthermore, DCs and not the tumor cells represent the major immune suppressive component in the OSCC ME. Thus, if really nothing change in the coagulome of OSCC cells upon treatment, the author should show that coagulome variations occurs in ME cells (e.g. fibroblast and macrophages) and ICP alterations in DCs.

This is again an important point. Indeed, we are reporting an analysis of the coagulome of the main cell types that constitute the TME. We anticipate that any change in tumor composition that would be induced for example by drug treatment would shape the coagulome in the long term. Importantly however, such changes cannot be easily studied in vitro. Once again, in the revised manuscript, we have rephrased the discussion to avoid any speculation on the stability of the coagulome. Clearly, more studies are required to address this difficult question.

14/ Altogether, this paper offer a broad descriptive analysis of OSCC coagulome gene expression based on TCGA and previous publish data. At this point, it is acceptable and in line with the title, offering an overall idea of the oral squamous cell carcinoma coagulome. Nevertheless, the last paragraph appears much weaker than the others, making drop the overall impact and quality of the paper.

Once again, we would like to thank the reviewer for carefully reading and commenting our manuscript. We do appreciate the useful comments and suggestions that we have received. Our reply regarding the last paragraph of the study can be found in detail in points #2 and #11 of this letter. We have worked on the last paragraph to improve its link to the rest of the study.

Round 2

Reviewer 1 Report

The author rapidly answered to my corners and questions. The overall quality of the paper has been improved.